# Newborn body composition after maternal bariatric surgery

Emma Malchau Carlsen[1]*, Kristina Martha Renault[2,3], Bertha Kanijo Møller[1], Kirsten Nørgaard[4], Jens-Erik Beck Jensen[5], Jeannet Lauenborg[6], Dina Cortes[1,7], Ole Pryds[1,8]

**1** Department of Pediatrics, Copenhagen University Hospital Hvidovre, Hvidovre, Denmark, **2** Department of Obstetrics and Gynaecology, Copenhagen University Hospital Hvidovre, Hvidovre, Denmark, **3** Obstetric Clinic, Copenhagen University Hospital Rigshospitalet, Copenhagen, Denmark, **4** Steno Diabetes Center Copenhagen, Gentofte, Denmark, **5** Department of Endocrinology, Copenhagen University Hospital Hvidovre, Hvidovre, Denmark, **6** Department of Obstetrics and Gynaecology, Copenhagen University Hospital Herlev, Herlev, Denmark, **7** Faculty of Health and Medical Science, University of Copenhagen, Copenhagen, Denmark, **8** Department of Pediatrics, Randers University Hospital, Randers, Denmark

* emc@dadlnet.dk

## Abstract

### Introduction

In pregnancy after Roux-en-Y gastric bypass (RYGB), there is increased risk of low birth-weight in the offspring. The present study examined how offspring body composition was affected by RYGB.

### Material and methods

Mother-newborn dyads, where the mothers had undergone RYGB were included. Main outcome measure was neonatal body composition. Neonatal body composition was assessed by dual-energy X-ray absorptiometry scanning (DXA) within 48 hours after birth. In a statistical model offspring born after RYGB were compared with a reference material of offspring and analyses were made to estimate the effect of maternal pre-pregnancy body mass index (BMI), gestational weight gain, parity, gestational age at birth and newborn sex on newborn body composition. Analyses were made to estimate the impact of maternal weight loss before pregnancy and of other effects of bariatric surgery respectively. The study was performed at a university hospital between October 2012 and December 2013.

### Results

We included 25 mother-newborn dyads where the mothers had undergone RYGB and compared them to a reference material of 311 mother-newborn dyads with comparable pre-pregnancy BMI. Offspring born by mothers after RYGB had lower birthweight (335g, p<0.001), fat-free mass (268g, p<0.001) and fat% (2.8%, p<0.001) compared with reference material. Only 2% of the average reduction in newborn fat free mass could be attributed to maternal pre-pregnancy weight loss whereas other effects of RYGB accounted for 98%. Regarding reduction in fat mass 52% was attributed to weight loss and 47% to other effects of surgery.

**Data Availability Statement:** Data can not be shared publicly because of danish laws. Data can be available for those interested after retrieving permission from the Danish Institutional Data Access Board and Ethics Committee Capital Region

of Denmark if the researchers meet the criteria to access confidential data. The study participants must also consent before the data can be accessed. Contact the Danish Institutional Data Access Board, email: videnscenterfordataanmeldelser. rigshospitalet@regionh.dk Ethics Committee Capital Region of Denmark, email: vek@regionh.dk

**Funding:** The study was funded by the Hvidovre University Hospital and the funding had no role in study design, data analysis, decision to publish or preparation of the manuscript.

**Competing interests:** No authors have competing interests.

## Conclusion

Offspring born after maternal bariatric surgery, had lower birthweight, fat-free mass and fat percentage when compared with a reference material. RYGB itself and not the pre-pregnancy weight loss seems to have had the greatest impact on fetal growth.

## Introduction

Maternal obesity has both short- and long-term consequences for the offspring and increases the offspring's risk of developing overweight and obesity [1–6]. Once established, severe obesity is difficult to treat. Roux-en-Y gastric bypass (RYGB) has been proven to be an effective way to reverse or lower the degree of obesity and the related morbidity and mortality [7, 8]. Women of child-bearing age also undergo bariatric surgery [9].

Birthweight and newborn body composition are affected by maternal BMI and gestational weight gain (GWG), and both maternal obesity and excessive weight gain is associated with a higher birthweight and higher fat mass in the offspring [10–12]. We have previously shown that infants of obese mothers have higher fat mass at birth and an abdominal fat accumulation compared with infants of normal weight women. Furthermore, low birthweight was associated with a lower crude abdominal fat mass, but a higher proportion of fat mass placed abdominally [13].

The adverse body composition tracks into childhood and may contribute to obese women's offspring increased risk of developing overweight and obesity [4, 14]. Interventional programs which aimed to limit the GWG in obese women did not seem to affect offspring weight and body composition at birth [15].

Women who undergo bariatric surgery before pregnancy have a lower risk of developing gestational diabetes mellitus, lower offspring birthweight and lower risk of large for gestational age (LGA) infants, but a higher risk of small for gestational age (SGA) infants than weight matched non-operated women. In contrast the risk of preterm delivery and having SGA infants is higher. Some studies have also found that perinatal mortality may be increased in pregnancy after RYGB [16–19]. There are no studies on how newborn body composition is affected by maternal bariatric surgery.

The aim of the present study was to examine how term newborn infant total and abdominal body composition is affected by maternal bariatric surgery, with the hypothesis that offspring born after maternal bariatric surgery have a lower total fat mass, and that they accumulate a relative higher proportion of their fat tissue abdominally. We also wanted to examine the impact of bariatric surgery on offspring birthweight and if body composition of the newborn was associated with maternal weight loss before pregnancy or surgery induced metabolic changes.

## Material and methods

Pregnant women with previous RYGB and their offspring were included consecutively between the 2nd of October 2012 and the 1st of December 2013 at Copenhagen University Hospital Hvidovre, Denmark, with more than 7000 deliveries annually. All term singleton, RYGB pregnancies were offered inclusion, there were no exclusion criteria other than maternal chronic disease such as insulin dependent diabetes or pre-eclampsia. A mixed group of 80 normal weight and 231 obese mother-newborn dyads was used as reference material. None of the women in the reference material had undergone RYGB or other bariatric surgery. We

consecutively recruited 80 singletons, healthy infants born at term (>258 days of gestation) and their mothers within 48 hours after delivery. The obese mothers had participated in the Treatment of Obese Pregnant (TOP) study at Copenhagen University Hospital Hvidovre [15]. In this subset of the TOP-study cohort the intervention did neither affect birthweight nor newborn body composition [13]. *Exclusion criteria*: Mothers with a chronic disease or pre-eclampsia were excluded. Infants requiring admission to neonatal intensive care unit or suffering from congenital diseases were also excluded. Furthermore, children born prematurely were excluded.

## Maternal and infant data

Information on parity, maternal social and smoking status, exercise habits was collected from a questionnaire filled in during the first trimester survey. For women with previous RYGB the date for the bariatric surgery, preoperative weight and the weight loss before pregnancy onset was registered.

Pre-pregnancy weight was self-reported. All women were weighed at gestational week 36–37 at the hospital wearing light clothing and no shoes (Seca digital scales, Seca, Germany). GWG was calculated as the difference between pre-pregnancy weight and weight at 36–37 weeks of gestation. Height was collected through hospitals files at inclusion. Pre-pregnancy body mass index (BMI) was calculated using the self-reported weight. Gestational age was determined by ultrasound at the nuchal transluciency scan in week 11 to 14.

The infants were weighed recumbent (Seca 727, digital baby scales, Seca, Germany). Length and head circumference were measured with non-stretchable measuring tape according to WHO guidelines [20]. Abdominal circumference was measured by non-stretchable measuring tape during mid-expiration at umbilical level in the supine position. Size for gestational age was calculated based on the reference from Marsal et al. [21].

## Body composition assessment of the newborn offspring

Within 48 hours after birth, newborn body composition was assessed using DXA scanning (DXA, Hologic 4500, Bedford, MA, USA). This method calculates fat and fat-free body mass as well as bone mass. Total X-ray dose for a whole-body scan was 10.5 µSv, equivalent to 1 to 2 days of background radiation. We used the same criteria as Cooke et al. when evaluating the scans, and only scans that met predefined quality criteria were included in the analysis [22]. Fat (%) was calculated as fat mass/total mass.

To estimate abdominal fat mass and fat-free mass, two abdominal regions were identified using the paediatric DXA software. The thoracic diaphragm and both upper iliac crests limited the first region, while the second region was limited laterally by the upper iliac crests and caudally by the femoral heads. The sum of the two regions constitutes the abdominal region. Abdominal/total fat ratio was calculated as abdominal fat mass/total fat mass. From duplicate scans in 58 infants, the test-retest variability concerning DXA-derived fat and fat-free mass was calculated to 11.8% and 7.1%, respectively [23].

## Ethics

The Ethics Committee of the Capital Region of Denmark (H-D-2008-119) approved the study, and written informed consent was obtained from both parents before the mother and infant were included in the study.

## Statistical analysis

Newborn offspring of women with previous RYGB where compared with a reference material consisting of a mixed group of obese and normal weight women's infants. Means and standard deviations (SD) were calculated for all normally distributed outcome variables. Data were compared by Student's *t*-test. Differences in proportions were tested using the Chi-square test. Simple linear regression analyses were also performed, where the effect of bariatric surgery (as a dichotomous outcome) were evaluated. In the analyses infant birth weight, fat-free mass and fat percentage were used as dependent variables. Analyses were adjusted for maternal age, smoking, primiparity, pre-pregnancy BMI, GWG, and infant sex and gestational age at birth.

In order to explore the effect of RYGB on foetal growth, body composition compartments in newborns of mothers with RYGB were compared with expected values. The expected values were calculated from the previous derived regression coefficients based on data in the reference material [13]. We used the reference material to calculate estimates of expected values of birthweight and each body composition compartment in infants of women with RYGB. For each infant, we made two sets of estimated values, one based on the maternal pre-surgery BMI and one based on the post-surgery, pre-pregnancy BMI. Furthermore GWG, parity, infant sex and gestational age at birth were incorporated in the calculations. These estimates were compared with actual measured values by means of paired t-test.

To illustrate our method, we present, as example, the regression used to calculate expected birthweight:

$$\text{Expected birthweight} = (-3086.1) + (\text{pre–preganancy BMI or pre–surgery BMI} * 13.8) + (\text{GWG} * 22.9) + (\text{days of gestation at birth} * 22.3) + (\text{if primiparity} - 192) + (\text{if female sex} - 63.8).$$

The impact of weight loss and RYGB on body composition in compartments in percent, were derived from the regressions models. The difference in the compartments between pre-surgery and pre-pregnacy BMI were compared to actual measured values. The difference that could not be explained by the change in pre-pregnancy BMI, was analysed as it derived from changes induced by bariatric surgery.

P values < 0.05 were set as significant (SPSS Statistics, version 19.0, Chicago, IL, USA).

## Results

In the study period, there was 53 women pregnant with previous bariatric surgery. We included 25 (47%) bariatric surgery mother-newborn dyads (Fig 1). Women who delivered at another hospital (n = 13), who gave birth prematurely (n = 6) and births where the infant was admitted to the neonatal intensive care unit (NICU) (n = 1) were excluded. Eight women declined participation. There were no differences in baseline maternal characteristics in women who gave birth premature and the case that was admitted to NICU (n = 7) when compared to included women (data not shown). No data were collected on the women that delivered at another hospital.

The reference material consisted of 311 mother-newborn dyads (n = 231 obese and n = 80 normal weight mothers). There were no differences in maternal characteristics including pre-pregnancy BMI and GWG when comparing women with previous RYGB with the reference material. However, there were large differences in newborn characteristics (Table 1).

Offspring born after bariatric surgery were in average 9.3% lighter, 2.3% shorter, had 2.6% smaller head circumference, and had 36.1% lower fat and 8% lower fat-free mass compared with the reference material. There were no differences in newborn sex and gestational age at

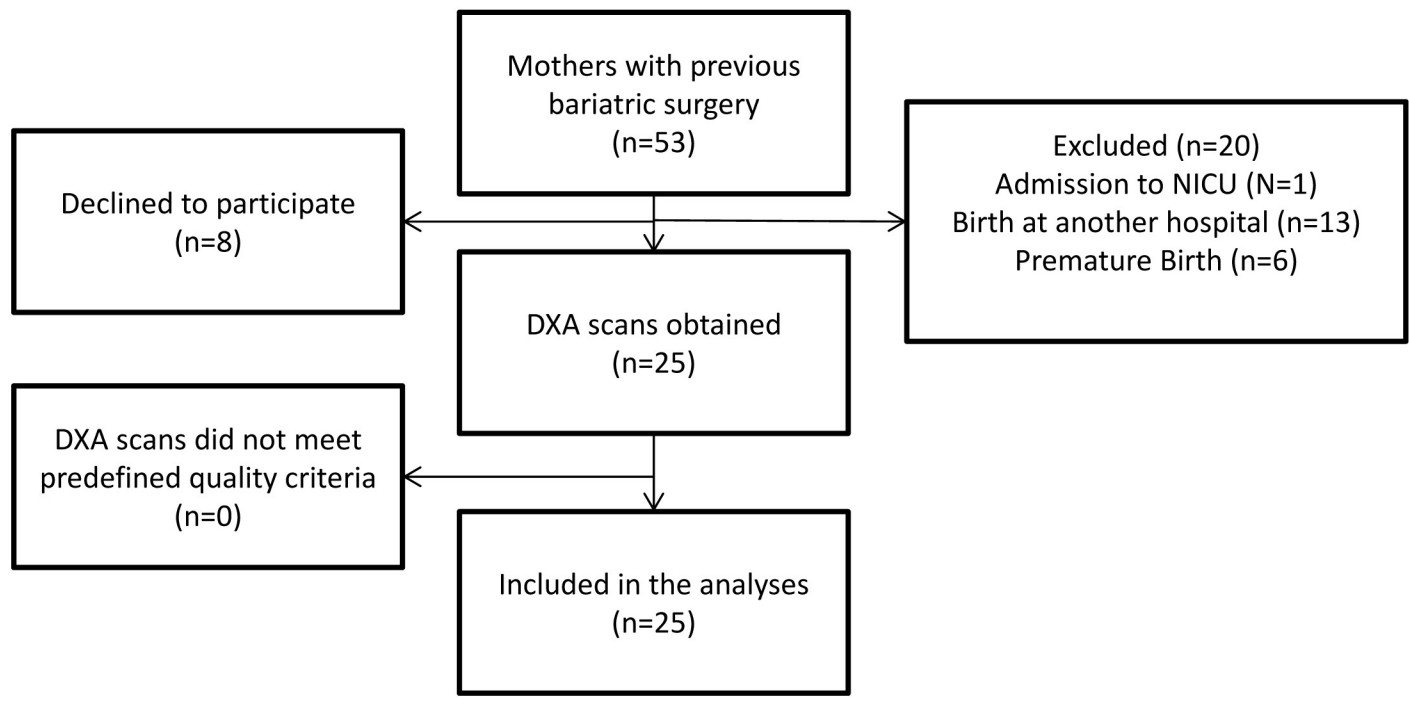

**Fig 1. Inclusion of women with previous bariatric surgery.**

birth (Table1). The effect of RYGB on birth weight (-298 g, p = 0.002), fat-free mass (-253 g, p = 0.001) and fat percentage (-2.5%, p = 0.002) persisted when performing linear regression, adjusted for factors known to influence infant growth; maternal age, smoking, pre-pregnancy BMI, GWG, primiparity, smoking, infant sex and gestational age at birth (Table 2).

The risk of preterm birth was higher in offspring born after RYGB (6/53) versus the reference material (8/404) (p<0.001).

### Estimated impact of maternal weight loss and of other effects of RYGB

Maternal RYGB before pregnancy resulted in an average reduction in fat free mass of 283 g (value estimated based on pre-surgery BMI 3345g- actual measured value 3062g) in the newborn offspring. Based on the previously reported regression coefficients of associations between maternal pre-pregnancy BMI and GWG and offspring weight, and fat distribution, we calculated that 2% (5g/283g) of the average reduction in newborn fat free mass could be attributed to maternal weight loss before pregnancy (difference between estimated pre-surgery BMI fat-free mass and pre-pregnancy BMI fat-free mass was 5 g). Thus, other effects of the bariatric surgery accounted for 98% (278g/283g) (Table 3).

Regarding reduction in fat mass of 276 g (value estimated based on pre-surgery BMI 536g-actual measured value 260g), 47% (131g/276g) was attributed to weight loss (difference between estimated pre-surgery BMI fat mass and pre-pregnancy BMI fat mass was 131 g). Other effects of RYGB lowered fat mass with 52% (145g/276g) (Table 3) (Fig 2a and 2b).

### Discussion

In the present study, we found that newborn offspring of mothers with RYGB before pregnancy had a lower birthweight, a lower fat-free mass, lower fat mass, lower fat percentage, and

**Table 1. Baseline characteristics for mothers with or without RYGB and for their newborns.**

| Maternal characteristics | Bariatric surgery mothers n = 25 | Reference material n = 311 | p-value |
|---|---|---|---|
| Maternal age (years)[1] | 30.3 (4.1) | 31.2 (4.6) | 0.35 |
| Pre-pregnancy BMI (kg/m$^2$) [1] | 28.8 (4.8) | 30.9 (6.0) | 0.07 |
| Gestational weight gain (kg) [1] | 13.2 (8.3) | 11.4 (6.1) | 0.15 |
| Primipara n (%)[2] | 13 (52) | 188 (60) | 0.41 |
| Maternal smoking (n) (%) | 4 (5.2) | 30 (9.6) | |
| Placental weight (g)[1] | 645 (91) | 665 (151) | 0.33 |
| Pre-surgery BMI (kg/m$^2$) [1] | 44.0 (5.4) | | |
| Weight loss after surgery until conception(kg) [1] | 40.8 (12.9) | | |
| Time from surgery to birth (months) [1] | 29.7 (11.6) | | |
| Newborn characteristics | | | |
| Birthweight (g) [1] | 3284 (327) | 3619 (523) | <0.001 |
| Birth length (cm) [1] | 51.0 (1.5) | 52.2 (2.3) | 0.01 |
| Head circumference (cm) [1] | 34.2 (1.6) | 35.1 (1.6) | 0.007 |
| Abdominal circumference (cm) [1] | 32.0 (2.2) | 33.4(2.2) | 0.002 |
| Sex [2] | | | |
| Male (n) (%) | 13 (52) | 161 (52) | |
| Female (n) (%) | 12 (48) | 150 (48) | 0.98 |
| Gestational age at birth (days) [1] | 277 (9) | 280 (9) | 0.14 |
| Birthweight Z-score * [1] | -0.45 (1.0) | 0.12 (1.10) | 0.01 |
| Size for gestational age(n)(%) * | | | |
| SGA | 1 (4) | 13 (4) | |
| AGA | 24 (96) | 276 (89) | |
| LGA | 0 (0) | 22 (7) | |
| Birthweight (n) [2] | | | |
| <2500 g | 0 | 8 | |
| 2500–4000 g | 24 | 229 | |
| >4000 g | 1 | 74 | 0.04 |
| Fat-free mass (g) [1] | 3062 (308) | 3330 (398) | <0.001 |
| Fat mass (g) [1] | 260 (126) | 407 (204) | < 0.001 |
| Fat (%) [1] | 7.7 (3.6) | 10.5 (4.3) | < 0.001 |
| Abdominal fat mass (g)[1] | 32 (17) | 53 (26) | < 0.001 |
| Abdominal/total fat mass (%)[1] | 11.2 (3.8) | 12.9 (3.2) | 0.02 |
| Body mass index (kg/m2) | 12.6 (1.3) | 13.2 (1.3) | 0.02 |

[1] Mean (±SD), Student's *t*-test.

[2] Proportion, chi-square test.

* Normalised birthweight adjusted for gestational age at birth and sex, according to Marsal et al. SGA (small for gestational age), AGA (appropriate for gestational age) and LGA (large for gestational age) [21].

a lower part of the fat distributed abdominally. The infants were shorter in length and had a smaller head circumference. The infants in the reference material were born to mothers with comparable pre-pregnancy BMI, parity, GWG and infant sex and gestational age, all factors known to determine infant birthweight and body composition [10, 11, 13].

We found that the effect of bariatric surgery was larger than that anticipated by weight loss alone. The pathophysiology behind the effect of bariatric surgery on birthweight and newborn body composition is not fully understood but possibly mediated through glucose metabolism, comprehensive hormonal alterations and altered absorption of nutrients from the intestines

**Table 2. Difference in infant birth weight, fat-free mass and fat (%) after bariatric surgery (RYGB) (n = 25), compared with reference material (n = 311).**

| Dependent variable: Newborn characteristics | RYGB infant, n = 25, Estimate (95% CI) | p-value |
|---|---|---|
| Birth weight (g) | -298 (-498–-106) | 0.002 |
| Fat-free mass (g) | -253 (-398–-109) | 0.001 |
| Fat (%) | -2.5 (-0.9–-4.1) | 0.002 |

Results are derived from linear regression. Estimate based on linear regression. Analyses adjusted for maternal age, smoking, pre-pregnancy BMI, GWG, primiparity, infant sex and gestational age at birth.

[6, 24, 25]. This reduces fetal growth. The consequences hereby are not known. It could, potentially, be harmful. Two small follow up studies have found that bariatric surgery does not induce long-term growth restriction [26, 27]. To our knowledge, there is only one study examining cognitive function in infants born after maternal bariatric surgery. This study observed that offspring born after bariatric surgery had impaired speech development. The authors conclude that this was not induced by bariatric surgery since the affected children were appropriate of size at birth [28]. There is no knowledge on long-term effects of the altered body-composition and fat-distribution. And future follow-up studies on the offspring are needed.

A higher proportion of offspring of obese women are LGA. There are benefits related to a normalized birthweight; a reduction of numbers of birth traumas and neonates with hypoglycaemia [1, 6].

A central question that remains unanswered is whether lowered birthweight and altered body composition, with lower total and abdominal fat-mass as well as lower fat percentage affects future risks for obesity. A large meta-analysis has shown that birthweight above 4000 g doubles the subsequent risk of becoming overweight and obese [29]. There are (as far as we are aware of) only two small studies examining (follow up 2–18 years) consequences of bariatric surgery on the offspring risk of developing overweight and obesity, in both studies siblings born before maternal bariatric surgery were used as reference. The results are contradictive.

**Table 3. Body composition in offspring of mothers after RYGB.**

| Newborn characteristics | RYGB mothers n = 25 | Expected values based on pre-pregnancy BMI | Difference between expected value based on pre-pregnancy BMI and measured value | Expected values based on pre-surgery BMI | Difference between expected value based on pre-surgery BMI and measured value |
|---|---|---|---|---|---|
| Birthweight | 3284 (374) | 3582 (262) | 299 (141–456)* | 3788 (270) | 504 (334–665)* |
| Fat-free mass (g) | 3062 (308) | 3340 (89) | 283 (161–405)** | 3345 (87) | 278 (171–385)* |
| Fat mass (g) | 260 (127) | 405 (73) | 145 (86–204)* | 536 (86) | 276 (212–340)* |
| Fat (%) | 7.7 (3.4) | 10.5 (1.4) | 2.7 (1.2–4.3)** | 13.4 (1.7) | 5.7 (4.1–7.3)* |
| Abdominal fat mass (g) | 32 (21) | 52 (9) | 20 (11–29)* | 70 (13) | 38 (8–68)* |

Actual measured values in first column, expected values based on maternal **pre-pregnancy BMI** in second column and expected values based on **pre-surgery BMI** in fourth column.

Data presented as mean (standard deviation) except for last column presented as mean (95% confidence interval).

Expected values were derived by linear regression using a reference material [13]. Calculated based on pre-pregnancy pre-surgery BMI. Parity, GWG, gestational age at birth and offspring sex were included in the analysis. The difference between the actual measured value and pre-pregnancy estimated value, was set as the effect induced by bariatric surgery.

Differences between the actual and estimated values compared using paired t-test.

*p-value <0.001,

**p-value = 0.001.

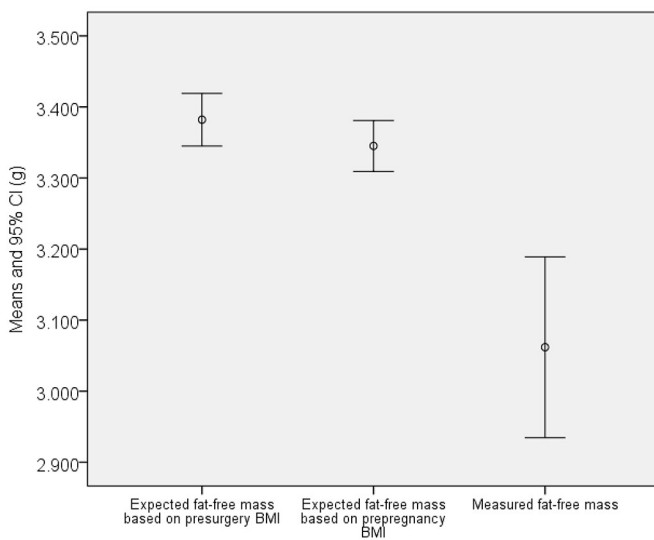
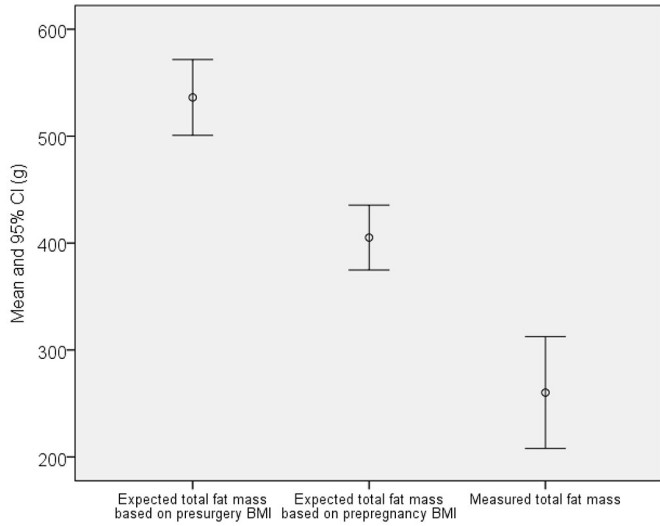

**Fig 2.** (a) Total fat mass in the newborn offspring: Expected values in gram calculated in the reference material using respectively presurgery BMI and prepregnancy BMI and measured values in the group where mothers had previous RYGB. (b) Total fat-free mass in the newborn offspring: Expected values in gram calculated in the reference material using respectively presurgery BMI and prepregnancy BMI and measured values in the group where mothers had previous RYGB.

One shows lowered risk for overweight and obesity (n = 45 sibling pairs), whereas the other finds no effect of bariatric surgery on offspring risk of becoming overweight and obese (n = 39 sibling pairs) [26, 27]. More and larger studies on this subject are warranted to be able to draw proper conclusions.

We have previously shown that infants with a low birthweight have a lower total fat mass, and tend to accumulate their fat mass abdominally [13]. The long-term consequences of this is not known, but it might be linked to the low birthweight infants' abdominal fat accumulation as children and adults, as well as their long-term increased risk of developing type II diabetes [30–32]. In this study, there were no sign of that the infants born after maternal RYGB accumulated fat mass abdominally. There was, however, only one SGA infant in the bariatric surgery group. We only included infants born at term. So potential pregnancies with severe fetal growth restriction and delivery before gestational age 37 might not be represented in our study. In a Danish study from 2013 there were 7.1% SGA infants among mothers with previous bariatric surgery, compared with 2.9% in the matched reference material. In our small study only one (4%) of the offspring of mothers with RYGB before pregnancy were SGA [16]. One should bear in mind that a possible beneficial effect of bariatric surgery on offspring risk of developing overweight and obesity can thus be replaced by an increased risk of morbidity related to being SGA at birth.

## Strengths and limitations

We only examined 25 term born infants born after bariatric surgery, but despite this, we found significant differences in both birthweight and body composition. Our findings should, however, be interpreted with caution due to the low number included. Our study was restricted to term born infants, and six were excluded due to premature birth. The risk of preterm birth was higher in pregnancies after bariatric surgery than in the reference material, and this relates well with previous findings. In this cohort there is a low proportion of SGA infants and this may influence the analyses on the effect of RYGB on newborn fat distribution [33, 34].

In our further analyses, we have chosen to use both maternal pre-pregnancy BMI and presurgery BMI. By using pre-pregnancy BMI we believe to estimate the gross metabolic effect of

bariatric surgery on newborn body composition. We also included an estimate of the effect of post-surgery weight loss in our analyses, by making an estimate of newborn infants body composition compartments based on maternal pre-surgery BMI. One should bear in mind that we used the same GWG in both pre-surgery and pre-pregnancy estimates of infants body composition compartments and we have not accounted for the fact that GWG is in numerous previous studies shown to be inversely related to pre-pregnancy BMI [12].

The reference material was partly recruited through an interventional program which aimed to lower GWG. The program had no effect on mean GWG, birthweight and newborn body composition in the subgroup included in our study. But the fact that the women were followed through pregnancy might have elevated their awareness regarding life-style issues, leading to a healthier life-style, which might lead to and underestimation of the effect of bariatric surgery on birthweight and body composition. The cohort as a whole was a rather homogenous group, which reduce the risk of bias, but lower the external validity of our findings.

## Conclusion

Offspring of mothers with RYGB before pregnancy born at term have lower birthweight, fat-free mass and fat percentage when compared with reference material. There is nothing that indicates aberrant fat deposition in this group of term born, infants. It seems like RYGB itself and not the pre-pregnancy weight loss had the greatest impact on offspring body composition. The long-term consequences of these findings need to be further examined.

## Acknowledgments

The authors would like to express their deepest gratitude to all participating families. All analyses this paper is based on, is available on request by contacting the corresponding author. Danish laws forbid us to publish the data the study was based on.

## Author Contributions

**Conceptualization:** Emma Malchau Carlsen, Kristina Martha Renault, Bertha Kanijo Møller, Kirsten Nørgaard, Jens-Erik Beck Jensen, Jeannet Lauenborg, Dina Cortes, Ole Pryds.

**Data curation:** Bertha Kanijo Møller, Kirsten Nørgaard, Jens-Erik Beck Jensen, Dina Cortes, Ole Pryds.

**Formal analysis:** Emma Malchau Carlsen, Kristina Martha Renault, Kirsten Nørgaard, Jens-Erik Beck Jensen, Jeannet Lauenborg, Ole Pryds.

**Funding acquisition:** Kristina Martha Renault, Jeannet Lauenborg, Ole Pryds.

**Investigation:** Emma Malchau Carlsen, Kristina Martha Renault, Bertha Kanijo Møller, Kirsten Nørgaard, Jens-Erik Beck Jensen, Jeannet Lauenborg, Dina Cortes, Ole Pryds.

**Methodology:** Emma Malchau Carlsen, Kristina Martha Renault, Bertha Kanijo Møller, Jens-Erik Beck Jensen, Jeannet Lauenborg, Dina Cortes, Ole Pryds.

**Project administration:** Emma Malchau Carlsen, Kristina Martha Renault, Bertha Kanijo Møller, Dina Cortes, Ole Pryds.

**Resources:** Jeannet Lauenborg, Ole Pryds.

**Supervision:** Kristina Martha Renault, Bertha Kanijo Møller, Kirsten Nørgaard, Jens-Erik Beck Jensen, Jeannet Lauenborg, Dina Cortes, Ole Pryds.

**Validation:** Emma Malchau Carlsen, Ole Pryds.

**Visualization:** Ole Pryds.

**Writing – original draft:** Emma Malchau Carlsen, Kristina Martha Renault, Bertha Kanijo Møller, Kirsten Nørgaard, Jens-Erik Beck Jensen, Jeannet Lauenborg, Dina Cortes, Ole Pryds.

**Writing – review & editing:** Emma Malchau Carlsen, Kristina Martha Renault, Bertha Kanijo Møller, Kirsten Nørgaard, Jens-Erik Beck Jensen, Jeannet Lauenborg, Dina Cortes, Ole Pryds.

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
