## [Decision Letter · Decision Letter 0]

25 Nov 2019

PONE-D-19-29375

Newborn body composition is severely affected by maternal bariatric surgery

PLOS ONE

Dear Dr Carlsen,

Thank you for submitting your manuscript to PLOS ONE. After careful consideration, we feel that it has merit but does not fully meet PLOS ONE’s publication criteria as it currently stands. Therefore, we invite you to submit a revised version of the manuscript that addresses the points raised during the review process.

We would appreciate receiving your revised manuscript by Jan 09 2020 11:59PM. To enhance the reproducibility of your results, we recommend that if applicable you deposit your laboratory protocols in protocols.io, where a protocol can be assigned its own identifier (DOI) such that it can be cited independently in the future. For instructions see: http://journals.plos.org/plosone/s/submission-guidelines#loc-laboratory-protocols

We look forward to receiving your revised manuscript.

Kind regards,

Andreas Beyerlein

Academic Editor

PLOS ONE

Journal Requirements:

1. Thank you for including your ethics statement:  "The Ethics Committee of the Capital Region of Denmark (H-D-2008-119) approved the study, and written informed consent was obtained from both parents before infants were included in the study".   

For studies reporting research involving human participants, PLOS ONE requires authors to confirm that this specific study was reviewed and approved by an institutional review board (ethics committee) before the study began.

2. Please provide additional details regarding participant consent. Specifically, please also include whether consent was obtained from the mothers enrolled in the study (women who underwent bariatric surgery, obese women and normal weight women). In the ethics statement in the Methods and online submission information, please ensure that you have specified (1) whether consent was suitably informed and (2) what type you obtained (for instance, written or verbal). If your study included minors under age 18, state whether you obtained consent from parents or guardians. If the need for consent was waived by the ethics committee, please include this information.

3. We noticed you have some minor occurrence(s) of overlapping text with the following previous publication(s), which needs to be addressed:

https://doi.org/10.1111/apa.12713

In your revision ensure you cite all your sources (including your own works), and quote or rephrase any duplicated text outside the Methods section. Further consideration is dependent on these concerns being addressed.

4. Please include additional information regarding the questionnaire used in the study and ensure that you have provided sufficient details that others could replicate the analyses. For instance, if you developed a questionnaire as part of this study and it is not under a copyright more restrictive than CC-BY, please include a copy, in both the original language and English, as Supporting Information.

Additional Editor Comments (if provided):

General:

- I understand that the data cannot be made publicly available. However, at least the analysis code could be put online, e.g. at www.osf.io, with the exact site being mentioned in the manuscript.

- Please provide a STROBE statement and adjust the manuscript accordingly where necessary.

Methods:

- Were there any exclusion criteria for the RYGB group regarding e.g. maximum time between RYGB and birth or intermittent pregnancies?

- Was it checked whether any of the reference mothers had had RYGB previously?

- How were SGA and LGA calculated?

- l.140-152: This approach is quite unusual. Why did the authors not simply calculate regression models on all mothers and report the regression coefficient for RYGB as a binary variable?

Results:

- Figure 1 should be described in more detail in the main text. Did the RYGB mothers who were excluded differ from those included in certain characteristics?

- Figure legends should be presented below the figures, not in the main text.

- Table 1: Suggest to provide all p-values, also the NS ones. The authors should report n(%) for all categorical variables. The chi-square test is not appropriate for maternal smoking, SGA/AGA/LGA and birth weight due to too low expected cell numbers.

- Table 2: Please provide 95% confidence intervals and state more clearly how the p-values were calculated.

Discussion:

- l.183/184: The preterm rate in the control children seemed unusually low. What could be a potential explanation for this low rate?

- l.265-268: Could the exclusion of the RYGB preterm children have biased the main findings? It may be worth to include them in a sensitivity analysis which could also be adjusted for preterm birth.

- Conclusion (and title): Wording implying causation should be avoided in cross-sectional studies.

Reviewers' comments:

Reviewer's Responses to Questions

**Comments to the Author**

1. Is the manuscript technically sound, and do the data support the conclusions?

Reviewer #1: Yes

Reviewer #2: Partly

2. Has the statistical analysis been performed appropriately and rigorously? 

Reviewer #1: Yes

Reviewer #2: I Don't Know

3. Have the authors made all data underlying the findings in their manuscript fully available?

Reviewer #1: No

Reviewer #2: No

4. Is the manuscript presented in an intelligible fashion and written in standard English?

Reviewer #1: Yes

Reviewer #2: Yes

5. Review Comments to the Author

Reviewer #1: ¬¬¬Review of manuscript - PONE-D-19-29375

Thank you for the opportunity to review this manuscript. This is a Danish study on offspring body composition after bariatric surgery compared with control births. In general, this is a very interesting and well conducted study of importance for the research field.

General comments

1. How was the model for expected values of birth weight developed? Does it use the Marsal curves as reference? Is it appropriate to use gestational weight gain in the model given that this could be a mediator between RYBG and the outcome under study? Could the model be overfitted?

2. The study included both obese and normal weight comparators. Is there a risk that the comparison would not be optimal? Would a matched analysis be superior where women of the same BMI would be compared?

3. How was the attribution of the average reduction in new-born free fat mass calculated?

4. Almost half of the study participants were excluded and did not perform the DXA analysis. Did they differ in maternal and neonatal characteristics including birth weight for gestational age? Could this have influenced the findings?

5. Were there any missing data for the variables included in the study? If yes, how was this handled in the analysis?

Specific comments

1. Please include proportions (%) in Table 1 for size for gestational age and birthweight.

2. In Table 1, was it weight loss from surgery to conception or birth? Please include this for the variable.

3. In Table 2, include as a footnote how the expected measures were obtained.

4. Please exclude paired t-test from the title of Table 2.

5. Could not early pregnancy weight be retrieved from the antenatal medical record instead of having it self-reported?

6. The discussion on the biological reason for the results could be further developed. What metabolic, nutritional and hormonal differences could induce such an effect?

Reviewer #2: In this manuscript, Malchau Carlsen et al. compared newborn's body composition measured by DXA within 48 h of birth in neonates born after maternal bariatric surgery (RYGB) (N=25) with measures obtained from a reference sample of newborns from obese and normal-weight mothers (similar pre-pregnancy BMI). They found that neonates born after maternal RYGB had a lower birthweight, a lower fat-free mass, lower fat mass, lower fat percentage, and a lower part of the fat distributed abdominally than neonates from the reference sample.

This study reports novel data, and answers a clinically relevant question, that has not been assessed previously in litterature. Whereas the sample size is rather small (N=25) for the post- bariatric surgery sample, the study uses standard measurements of newborn's body composition.

Major:

1. The selection of the "reference material" is not clearly explained. Some details are not described in the method and are important for comprehension and interpretation of the data: How the mothers in reference sample were recruited and how mothers included in this specific analysis were selected ? Were the procedures exactly the same in the two studies? Was the reference cohort matched for any characteristics of the RYGB cohort?

2. I suggest that the statistical analysis should be reviewed by a statistician, as I am not in measure to assess the quality of the analysis proposed. The interpretation of the results of the expected body fat mass based on previously described associations as a % contribution from each variable needs to be commented by a statistician as it may not be appropriate to interpret directly as the contribution of a variable to the outcome. As this is an important message from the manuscript, interpretation of these models should be done with caution.

3. The authors should explain the sample size determination for both groups.

Minor:

Line 153: RYGB instead of RYGBP

Line 199-200: these interpretation of data should be consider in the discussion rather than results section.

Line 220-223: a reference for glucose metabolism and hormonal alteration's impact on body fat mass would be appreciated

6. PLOS authors have the option to publish the peer review history of their article (what does this mean?). If published, this will include your full peer review and any attached files.

Reviewer #1: Yes: Olof Stephansson

Reviewer #2: No

---

## [Author Response · Author response to Decision Letter 0]

8 Feb 2020

Editors comments:

General: - I understand that the data cannot be made publicly available. However, at least the analysis code could be put online, e.g. at www.osf.io, with the exact site being mentioned in the manuscript.

First of all: Thank you for a detailed review of our manuscript that certainly improved the quality of our work. 

We are, unfortunately not, familiar with the website mentioned. The analysis will be available by contacting the corresponding author. Please, refer to comment under acknowledgement.

 - Please provide a STROBE statement and adjust the manuscript accordingly where necessary. Strobe statement is added, and the checklist is attached (Appendix 1).

 Methods: - Were there any exclusion criteria for the RYGB group regarding e.g. maximum time between RYGB and birth or intermittent pregnancies?

No exclusions, all pregnancies were included after informed oral and written consent was retrieved.

 - Was it checked whether any of the reference mothers had had RYGB previously?

Yes, there was no woman in the reference material that had undergone bariatric surgery of any kind, it has been added to the methods section.

 - How were SGA and LGA calculated?

Sorry for not making this clear. We used Marsal’s reference, which is the reference used by Danish neonatologists (and used in most Nordic countries). This is more clearly stated in the manuscript now, please see line 118-119.

 - l.140-152: This approach is quite unusual. Why did the authors not simply calculate regression models on all mothers and report the regression coefficient for RYGB as a binary variable? We chose this approach since it was the only way we could estimate the effect of weight loss induced by surgery and the effect of bariatric surgery on foetal growth during pregnancy. 

A linear regression, with RYGB as a binary variable, has been calculated and added as Appendix 2 to this review and is of course also informative. 

We, however, believe that the analyses we made are better to illustrate the enormous effect bariatric surgery has on fetal growth since it is this and not the weight loss that induce the primary changes. We find it interesting to see that the women that had undergone bariatric surgery has the same mean gestational weight gain and there is also no difference I placental weight. Despite this, their offspring is significantly smaller.

 Results: - Figure 1 should be described in more detail in the main text. Did the RYGB mothers who were excluded differ from those included in certain characteristics?

We did not collect data on the women who gave birth at another hospital (n=13). There was no difference in baseline maternal characteristic in the women that gave birth premature or the got admitted to NICU (n=7). This also added to the text.

 - Figure legends should be presented below the figures, not in the main text.

It has been changed.

 - Table 1: Suggest to provide all p-values, also the NS ones. The authors should report n(%) for all categorical variables. The chi-square test is not appropriate for maternal smoking, SGA/AGA/LGA and birth weight due to too low expected cell numbers.

All p-values are now added as well as n(%) for all categorical values. The Chi-square test for maternal smoking, SGA/AGA/LGA and birthweight has been removed. 

 - Table 2: Please provide 95% confidence intervals and state more clearly how the p-values were calculated.

The 95 CI% and SD are added. We have tried to clarify how the p-values were calculated.  Discussion: - l.183/184: The preterm rate in the control children seemed unusually low. What could be a potential explanation for this low rate?

We cannot explain the low rate of prematurity in the control children and we were also surprised by this finding.

 - l.265-268: Could the exclusion of the RYGB preterm children have biased the main findings? It may be worth to include them in a sensitivity analysis which could also be adjusted for preterm birth.

We did not dexa-scan the premature born children. The incidence of growth restriction is significantly higher in the premature born babies and our aim was to examine how term newborn body composition was affected by bariatric surgery. Therefore, women with pre-eclampsia were also excluded.

 - Conclusion (and title): Wording implying causation should be avoided in cross-sectional studies.

Sorry, the title has been changed and the wording in the conclusion has been changed.

Reviewer #1:  Thank you for the opportunity to review this manuscript. This is a Danish study on offspring body composition after bariatric surgery compared with control births. In general, this is a very interesting and well conducted study of importance for the research field. General comments

Thank you for reviewing our article, which improved our work. 1. How was the model for expected values of birth weight developed? Does it use the Marsal curves as reference? Is it appropriate to use gestational weight gain in the model given that this could be a mediator between RYBG and the outcome under study? Could the model be overfitted?

We calculated expected values using our reference material. We used linear regression. For each woman, expected values for birth weight and body composition compartments were calculated based on the pre-surgery and pre-pregnancy BMI. The expected value was compared to the actual measured value. We used the actual measured gestational weight gain (GWG).

As an example, we show you the regression used to calculate birth weight (this is also presented in the statistics section):

Expected birthweight = (- 3086.1)+ (pre-preganancy BMI or pre-surgery BMI*13.8)+(GWG*22.9)+(days of gestation at birth*22.3)+(if primiparity -192)+(if female sex -63.8).

It is important to notice that there was no difference in mean GWG comparing the RYGB women with the reference material and we do not believe this mediate the change in birth weight and newborn body composition since the RYGB women delivered smaller babies despite an appropriate mean GWG. One should also notice that there is no difference in mean placental weight.

We have tried to clarify our method under the statistics section.

 2. The study included both obese and normal weight comparators. Is there a risk that the comparison would not be optimal? Would a matched analysis be superior where women of the same BMI would be compared?

We considered a matched analysis when we started the work. We believe the method used is appropriate, using linear regression we calculate an expected outcome in the reference material based on BMI, GWG, gestational age at birth, parity and newborn sex. The problems with a matched analysis would be to obtain enough controls with high pre-pregnancy BMI, few women have pre-pregnancy BMI of 44, uncomplicated pregnancy and uneventful delivery. The mean pre-surgery BMI was 44.0.

3. How was the attribution of the average reduction in new-born free fat mass calculated?

We have tried to explain this in the statistics section and clarify it in the results section. Sorry for not doing it properly. The difference in the compartments between pre-surgery and pre-pregnacy BMI were compared to actual measured values. We concluded that the difference between the actual measured value and the estimated value based on the pre-pregnancy BMI was induced by the effect of RYGB during pregnancy. The babies are smaller and leaner than expected by the maternal pre-pregnancy BMI and GWG.

 4. Almost half of the study participants were excluded and did not perform the DXA analysis. Did they differ in maternal and neonatal characteristics including birth weight for gestational age? Could this have influenced the findings?

Please see the answer to the editors question above. We did not obtain neonatal characteristics on excluded infants. We would guess that a proportion of babies delivered premature have lower birth weight SDS. Including them, might have done the differences more pronounced. 

 5. Were there any missing data for the variables included in the study? If yes, how was this handled in the analysis?

We only included women for whom the offspring had a successful dexa-scan. The data are therefore complete. 

 Specific comments 1. Please include proportions (%) in Table 1 for size for gestational age and birthweight.

Proportions is now included.

 2. In Table 1, was it weight loss from surgery to conception or birth? Please include this for the variable.

It was until conception, and this is now included.

3. In Table 2, include as a footnote how the expected measures were obtained.

The footnote now include how expected values were obtained.

 4. Please exclude paired t-test from the title of Table 2.

Paired t-test is removed from title and added to footnote.

 5. Could not early pregnancy weight be retrieved from the antenatal medical record instead of having it self-reported?

In Denmark women do not visit their midwife/doctor until gestational week 8-10, therefore we have self-reported weight. This pre-pregnancy weight is also self-reported in the reference material.

 6. The discussion on the biological reason for the results could be further developed. What metabolic, nutritional and hormonal differences could induce such an effect? Based on current literature I am not sure we can further develop this, but if there are any reference we have not included that are relevant, we would like to do so.

 Reviewer #2: 

In this manuscript, Malchau Carlsen et al. compared newborn's body composition measured by DXA within 48 h of birth in neonates born after maternal bariatric surgery (RYGB) (N=25) with measures obtained from a reference sample of newborns from obese and normal-weight mothers (similar pre-pregnancy BMI). They found that neonates born after maternal RYGB had a lower birthweight, a lower fat-free mass, lower fat mass, lower fat percentage, and a lower part of the fat distributed abdominally than neonates from the reference sample. This study reports novel data, and answers a clinically relevant question, that has not been assessed previously in litterature. Whereas the sample size is rather small (N=25) for the post- bariatric surgery sample, the study uses standard measurements of newborn's body composition.

Thank you for considerate input, which have improved our paper.  

Major: 1. The selection of the "reference material" is not clearly explained. Some details are not described in the method and are important for comprehension and interpretation of the data: How the mothers in reference sample were recruited and how mothers included in this specific analysis were selected ? Were the procedures exactly the same in the two studies? Was the reference cohort matched for any characteristics of the RYGB cohort? The reference material was recruited in the same manner as the RYGB women. The reference cohort was not matched in any way. The reference material was recruited before the RYGB cohort. For further detail please read reference 13 and 15. 

 2. I suggest that the statistical analysis should be reviewed by a statistician, as I am not in measure to assess the quality of the analysis proposed. The interpretation of the results of the expected body fat mass based on previously described associations as a % contribution from each variable needs to be commented by a statistician as it may not be appropriate to interpret directly as the contribution of a variable to the outcome. As this is an important message from the manuscript, interpretation of these models should be done with caution. We have consulted the statistician at our hospital, but any external review would also be welcome. Please see answer to reviewer 1 first comment.

 3. The authors should explain the sample size determination for both groups.

We wanted to describe differences in birth weight and body composition after bariatric surgery, and there is no previous data on this, therefore sample size calculation was difficult.

 Minor: Line 153: RYGB instead of RYGBP Thank you, it has been changed.

Line 199-200: these interpretation of data should be consider in the discussion rather than results section.

It has been deleted.

 Line 220-223: a reference for glucose metabolism and hormonal alteration's impact on body fat mass would be appreciated

We have added two references.

---

## [Decision Letter · Decision Letter 1]

6 Mar 2020

PONE-D-19-29375R1

Newborn body composition after maternal bariatric surgery

PLOS ONE

Dear Dr Carlsen,

Thank you for submitting your manuscript to PLOS ONE. After careful consideration, we feel that it has merit but does not fully meet PLOS ONE’s publication criteria as it currently stands. Therefore, we invite you to submit a revised version of the manuscript that addresses the points raised during the review process.

We would appreciate receiving your revised manuscript by Apr 20 2020 11:59PM. To enhance the reproducibility of your results, we recommend that if applicable you deposit your laboratory protocols in protocols.io, where a protocol can be assigned its own identifier (DOI) such that it can be cited independently in the future. For instructions see: http://journals.plos.org/plosone/s/submission-guidelines#loc-laboratory-protocols

We look forward to receiving your revised manuscript.

Kind regards,

Andreas Beyerlein

Academic Editor

PLOS ONE

Additional Editor Comments (if provided):

The authors did well to revise their manuscript in response to most of my questions. However, a few issues remain:

- l.197-205: These results should be described in much more detail. In particular, the birthweight findings should be mentioned, and the calculations of how much impact the surgery vs. weight loss had should be explained more clearly. Besides, I do not understand the authors' conclusion that 2% of the reduction in fat free mass could be attributed to weight loss, because the estimated difference based on pre-surgery compared to pre-pregnancy BMI was 5 g lower and not higher.

- l.209: Also the third column shows 95% CIs.

- I am still in doubt the analyses presented in table 2 are sound. In particular, I am wondering whether the authors took the expected values as fixed, which would mean that the 95% CIs for the differences between expected and measured values did not take the uncertainty in the estimation of the expected values into account. The authors should clarify this and/or revise their analyses accordingly.

- The results from Appendix 2 should be presented in suitable fashion, i.e. as scientific tables with explanations, and the results should also be mentioned in the main text and set into context to the results from Table 2.

- The authors should provide a statement in the main text why the analysis code cannot be made publicly available.

Reviewers' comments:

Reviewer's Responses to Questions

**Comments to the Author**

1. If the authors have adequately addressed your comments raised in a previous round of review and you feel that this manuscript is now acceptable for publication, you may indicate that here to bypass the “Comments to the Author” section, enter your conflict of interest statement in the “Confidential to Editor” section, and submit your "Accept" recommendation.

Reviewer #1: All comments have been addressed

Reviewer #2: All comments have been addressed

2. Is the manuscript technically sound, and do the data support the conclusions?

Reviewer #1: Yes

Reviewer #2: Yes

3. Has the statistical analysis been performed appropriately and rigorously? 

Reviewer #1: Yes

Reviewer #2: Yes

4. Have the authors made all data underlying the findings in their manuscript fully available?

Reviewer #1: Yes

Reviewer #2: Yes

5. Is the manuscript presented in an intelligible fashion and written in standard English?

Reviewer #1: Yes

Reviewer #2: Yes

6. Review Comments to the Author

Reviewer #1: No additional comments, my comments have been addressed properly. The manuscript is in my opinion ready for publication.

Reviewer #2: (No Response)

7. PLOS authors have the option to publish the peer review history of their article (what does this mean?). If published, this will include your full peer review and any attached files.

Reviewer #1: Yes: Olof Stephansson

Reviewer #2: No

---

## [Author Response · Author response to Decision Letter 1]

24 Mar 2020

The authors did well to revise their manuscript in response to most of my questions. However, a few issues remain:

 - l.197-205: These results should be described in much more detail. In particular, the birthweight findings should be mentioned, and the calculations of how much impact the surgery vs. weight loss had should be explained more clearly. Besides, I do not understand the authors' conclusion that 2% of the reduction in fat free mass could be attributed to weight loss, because the estimated difference based on pre-surgery compared to pre-pregnancy BMI was 5 g lower and not higher.

Thank you for your very useful comments, we are sorry we haven’t made this clear during the first review. We have tried to make it clearer in the text and hope we now have improved the description.

We looked at the difference in our estimates. Pre-surgery BMI estimates for mean fat-free mass was 3345 g, pre-pregnancy BMI estimates for fat-free mass 3340 g, the difference is hence 5 g, the effect of weight loss before pregnancy. The difference between pre-surgery fat-free mass and actual measured fat-free mass (3345 g - 3062 g) is 283 g. Therefore, weight loss only accounts for a small reduction in fat-free 5 g/283 g = 2%. The effect of bariatric surgery during pregnancy (276g/283g) induced the greatest change.

 - l.209: Also the third column shows 95% CIs.

That is correct.

 - I am still in doubt the analyses presented in table 2 are sound. In particular, I am wondering whether the authors took the expected values as fixed, which would mean that the 95% CIs for the differences between expected and measured values did not take the uncertainty in the estimation of the expected values into account. The authors should clarify this and/or revise their analyses accordingly.

Good point. We did not use the values as fixed, so the uncertainty is included in the 95%CI, which also are rather wide. 

 - The results from Appendix 2 should be presented in suitable fashion, i.e. as scientific tables with explanations, and the results should also be mentioned in the main text and set into context to the results from Table 2.

The results from appendix 2 are incorporated in our text now, and presented in a new table 2. The old table 2 is renamed table 3. The findings in table 2 are in line with table 3 (difference between pre-pregnancy BMI values and actual measured values). 

 - The authors should provide a statement in the main text why the analysis code cannot be made publicly available. A short statement has been added. Please see line 320.

---

## [Editor Report · Decision Letter 2]

27 Mar 2020

Newborn body composition after maternal bariatric surgery

PONE-D-19-29375R2

Dear Dr. Carlsen,

We are pleased to inform you that your manuscript has been judged scientifically suitable for publication and will be formally accepted for publication once it complies with all outstanding technical requirements.

With kind regards,

Andreas Beyerlein

Academic Editor

PLOS ONE

Additional Editor Comments (optional):

Well done. Please take care that the minus signs in the 95% CIs in table 2 will not get lost in the during production.
---

## [Editor Report · Acceptance letter]

1 Apr 2020

PONE-D-19-29375R2 

Newborn body composition after maternal bariatric surgery 

Dear Dr. Carlsen:

I am pleased to inform you that your manuscript has been deemed suitable for publication in PLOS ONE. Congratulations! Your manuscript is now with our production department. 

With kind regards,

on behalf of

Dr. Andreas Beyerlein 

Academic Editor

PLOS ONE